# Exploring Amodiaquine’s Repurposing Potential in Breast Cancer Treatment—Assessment of In-Vitro Efficacy & Mechanism of Action

**DOI:** 10.3390/ijms231911455

**Published:** 2022-09-28

**Authors:** Vineela Parvathaneni, Rameswari Chilamakuri, Nishant S. Kulkarni, Nabeela F. Baig, Saurabh Agarwal, Vivek Gupta

**Affiliations:** Department of Pharmaceutical Sciences, College of Pharmacy and Health Sciences, St. John’s University, 8000 Utopia Parkway, Queens, NY 11439, USA

**Keywords:** breast cancer, drug repurposing, amodiaquine, apoptosis, autophagy, 3D spheroids

## Abstract

Due to the heterogeneity of breast cancer, current available treatment options are moderately effective at best. Hence, it is highly recommended to comprehend different subtypes, understand pathogenic mechanisms involved, and develop treatment modalities. The repurposing of an old FDA approved anti-malarial drug, amodiaquine (AQ) presents an outstanding opportunity to explore its efficacy in treating majority of breast cancer subtypes. Cytotoxicity, scratch assay, vasculogenic mimicry study, and clonogenic assay were employed to determine AQ’s ability to inhibit cell viability, cell migration, vascular formation, and colony growth. 3D Spheroid cell culture studies were performed to identify tumor growth inhibition potential of AQ in MCF-7 and MDAMB-231 cell lines. Apoptosis assays, cell cycle analysis, RT-qPCR assays, and Western blot studies were performed to determine AQ’s ability to induce apoptosis, cell cycle changes, gene expression changes, and induction of autophagy marker proteins. The results from in-vitro studies confirmed the potential of AQ as an anti-cancer drug. In different breast cancer cell lines tested, AQ significantly induces cytotoxicity, inhibit colony formation, inhibit cell migration, reduces 3D spheroid volume, induces apoptosis, blocks cell cycle progression, inhibit expression of cancer related genes, and induces LC3BII protein to inhibit autophagy. Our results demonstrate that amodiaquine is a promising drug to repurpose for breast cancer treatment, which needs numerous efforts from further studies.

## 1. Introduction

Breast cancer is one of the most common type of cancers among women, making up to 25% of all new cancer diagnoses with continuously increasing incidence rates [1,2]. According to Susan G Komen, approximately 290,560 new cases of breast cancer will be diagnosed in the US, with >99% cases diagnosed in women. In addition, approximately 44,000 deaths will be reported in 2022 due to breast cancer, with a mortality rate of 19.4 per 100,000 [3]. Breast cancer is a heterogenous and complex disease, and is primarily caused by malignant lesions in the ductal epithelium of the breast [1]. Classification of different types of breast cancers into various categories has been widely published in the literature, where they are categorized based on the expression of certain important receptors such as estrogen receptor (ER), progesterone receptor (PR) and human epithelial receptor 2 (HER2) [4]. Different types of breast cancer include ER positive, PR positive, HER2 positive and triple negative breast cancer (TNBC) [5,6]. All these cell lines possess different risk factors for incidence, therapeutic response and distinct molecular features, thus reflect heterogenicity of the corresponding tumors [7,8]. Out of various categories, TNBC, i.e., tumor cells not expressing any of these ER/PR/HER2 receptors are considered to be the most aggressive type of breast cancer phenotype [9]. 

With recent advancements in the cancer therapeutics, multiple treatment choices are available for breast cancer treatment based on the type and aggressiveness of the cancer that have resulted in overall increase in patient survival (90% 5-year survival rate, 77% for TNBC) [10,11]. While chemotherapy, hormone therapy, immunotherapy, radiotherapy and surgery being the common modalities for breast cancer [7], primary choice of treatment includes surgery where complete resection of the major tumor mass is carried out [1]. US FDA has approved several drugs including both small molecule and macromolecular antibody treatments such as Docetaxel, Palbociclib, Olaparib, Trastuzumab etc. for treating breast cancer. However, treatment costs, acquired resistance, and off-target toxicity are major deterrents in success of current therapy and patient compliance [9]. Multi-drug resistance imposes another major problem and thus limits the success of chemotherapeutic regimen (TNBC tumors are resistant to standard treatment therapy) [12,13]. Non-specificity of chemotherapeutics is posing another drawback towards cancer treatment [14]. Furthermore, breast cancer encompasses several groups of distinct diseases with diverse clinical features [8]. Due to clinical heterogeneity, currently available treatment options are further complicated and moderately effective at best. Hence, it is highly recommended to comprehend different subtypes, understand the pathogenic mechanisms involved, and develop treatment modalities capable of tackling multiple different subtypes of the disease. Hence, there is a dire need to discover or develop new and effective approaches for breast cancer treatment, which are safe, possess less off-target side effects and efficacious while accelerating the drug development process [9,15].

Developing a new drug is a lengthy and cumbersome process, and requires a lot of investment [16,17], not to mention significant development obstacles resulting in clinical stage drug failure and higher attrition rates due to safety or efficacy issues [9]. Drug repurposing, i.e., finding new uses for old clinically approved drugs for new indication provides an exciting avenue for expediting the drug development process [18]. Due to availability of complete safety, pharmacokinetic, pharmacodynamic and toxicity profiles for these drugs, recycling old drugs enables their successful repurposing with reduced failures [15,18]. The potential of drug repurposing approach has been extensively validated, especially during ongoing COVID-19 pandemic [19]. Several successfully clinically repositioned drugs for breast cancer include Methotrexate (original indication: Leukemia), Goserilin (original indication: Prostate cancer), Vinblastine (original indication: Hodgkin lymphoma) etc. However, there are certain challenges and concerns associated with drug repurposing for breast cancer therapy which require thorough consideration. For instance, the tumor heterogeneity, poorly defined molecular signatures, and poorly identified drug dosage provides a checkpoint to drug repurposing approach, by limiting the patient pool available for a specific cancer subtype. Hence, it is critical to search the new therapeutic strategies for certain very stringent and hard to treat various molecular subgroups of breast cancers [9].

Mechanism based repurposing of an old FDA approved drug presents an outstanding opportunity to explore its efficacy in treating breast cancer [16]. For instance, many anti-malarial drugs have shown significant potential in treatment of various cancers, and have also been tested in clinical trials [20,21]. While originally developed to interact with and stop progression of malaria parasite, many antimalarial drugs have the ability to interfere with important oncogenic pathways, such as Wnt/β-catenin, STAT3, and NF-kB along with the capability to modulate cell death pathways, thus mediating the anti-tumor effects of antimalarials [22] Chloroquine, primaquine, and mefloquine particularly have been investigated for the treatment of numerous types of cancers, both alone and in combination with chemotherapy [23,24,25] However, there were some drawbacks limiting their success as effective anti-cancer therapies. There were reports about the issues related to kidney and other organ injuries after chloroquine’s use with chemo and radiotherapy [26]. Inadequacy in anti-cancer efficacy of chloroquine in tumor models was explained by Pellegrini et al. [27] Recently, our group reported the repurposing potential of another antimalarial, amodiaquine (AQ) for cancer therapeutics by encapsulating AQ in polymeric nanoparticles for non-small cell lung cancer [28]. In addition, AQ has also been reported to cause autophagic-lysosomal and proliferative blockade in melanoma cells [29]. AQ affects the autophagic flux at a late stage thus inhibiting the fusion of the autophagosomes with the lysosomes and subsequent degradation of the autolysosome as reported earlier [29,30]. In the current study, we have attempted to establish and validate the efficacy of amodiaquine, 4-[(7-chloroquinolin-4-yl) amino]-2-[(diethylamino) methyl] phenol (AQ), an FDA approved anti-malarial compound against various breast cancer types.

In the project, we hypothesize that AQ exerts its anticancer efficacy in majority of breast cancer subtypes namely ER, PR positive breast cancer types; triple negative breast cancer (TNBC); and human epidermal growth factor receptor 2 positive (HER2+) breast cancer cell types due to its ability to induce apoptosis and to inhibit autophagy in various cancers. In this project, we aim to identify the breast cancer types responsive to AQ treatment and to explore the efficacy of amodiaquine against breast cancer through multitude of in-vitro and ex-vivo cell models.

## 2. Results

### 2.1. Cytotoxicity against Breast Cancer Cell Lines

We have recently demonstrated the anti-cancer effects of amodiaquine in non-small cell lung cancer cells [31]. We tested amodiaquine against triple negative breast cancer (MDAMB231, and BT549); HER2 positive (SKBR3), and ER/PR positive/HER2 negative (MCF7) cell lines; to compare between SKBR3/BT549 vs. MDAMB231/MCF7 [8]. 

As observed in cytotoxicity studies (Figure 1), amodiaquine exhibited significant cytotoxic potential in all breast cancer cell lines tested with an IC_50_ of 11.5 ± 6.5 µM (MCF-7), 8.2 ± 2.8 µM (MDAMB-231), 24.0 ± 2.2 µM (BT-549) and 16.0 ± 3.9 µM (SK-BR-3). The results have demonstrated the varied cytotoxicity potential of AQ against different breast cancer types where dose-dependent inhibition of cell proliferation was recorded. Lower IC_50_ values have been observed for AQ in case of MCF-7 and MDAMB-231 compared to BT-549 and SK-BR-3. The observed differences in cytotoxicity may be due to the distinct properties of different subtypes of breast cancer. AQ may have the capability to exert its anti-cancer efficacy via Estrogen and progesterone receptors predominantly as both MCF-7 and MDAMB-231 cell lines are HER2 negative. Hence, it is assumed that AQ possesses superior cytotoxic potential against certain breast cancer types.

### 2.2. Scratch Assay

Scratch assay is a well-established method to access cell-cell interaction, and cellular migration [23]. For these experiments, a fresh scratch in cellular monolayer was imaged over a period of 12 h following AQ treatment, with representative images shown in Figure 2A at 0- and 12-h following treatment in MDAMB-231 cells. After 12 h, % scratch closure was found to be 75.0 ± 13.2%, 62.9 ± 15.0%, 55.5 ± 6.4% and 33.4 ± 13.9% for control, AQ 7.5 μM, AQ 15 μM and AQ 25 μM, respectively, in case of MDAMB-231 (Figure 2B). As can be seen, AQ showed significantly better efficacy in inhibiting cellular migration following 12-h treatment, at concentrations of 15 and 25 μM. (Control vs. AQ 15Μm—*p* < 0.05, Control vs. AQ 25 μM—*p* < 0.01). From Figure 2A,B, it can be understood that scratches treated with control and AQ 7.5 μM (non-significant as compared to control) showed migration of cells and those treated with AQ (15 and 25 μM) showed significant inhibition of cellular migration or scratch closure. This indicates the dose-dependent anti-migratory efficacy of AQ with in breast cancer, thus reducing the tumor metastasis probability.

### 2.3. Vasculogenic Mimicry Assay

Vasculogenic mimicry assay was performed to evaluate the inhibitory effect of AQ on vascular network formation of aggressive MDAMB-231 breast cancer cell line. Formation of 3D channel-like networks which are representative of the initial stages of vasculogenic mimicry were photographed under an inverted microscope (10×) (Laxco, Mill Creek, WA, USA) after 8 h. Cells were observed by phase-control microscopy during after the incubation period. As shown in Figure 2C, visually significant differences in vascular network formation by MDAMB-231 cells in presence of varying concentrations of AQ were observed. MDA-MB-231 cells generated vascular patterns consisting of a tubular network in case of control group. Tubular structures started were clearly visible by phase microscopy after 8 h. In contrast, no or minimal tubular networks were observed following incubation with AQ 7.5 µM and AQ 15 µM. In case of AQ 25 µM treatment, no channels were detected. These results provide direct evidence that inhibition of vascular network formation could be a potential mechanism of action for AQ in inhibiting breast cancer cell proliferation.

### 2.4. Clonogenic Assay

As repopulation of residual tumor cells to form recurrences depends on the capacity of cells to reproduce themselves, the effect of AQ on clonogenic growth was tested in the colony formation assay [25]. AQ was evaluated for its long-term efficacy using clonogenic assay in two different breast cell lines, MCF-7 (HER2 negative) and MDAMB-231 (Triple negative). From representative images shown in Figure 3A, it can be illustrated that colony growth was significantly inhibited by AQ compared to control, in both MCF-7 and MDAMB-231 cell lines. After 48-h treatment period and 7-day incubation thereafter, % of colonies survived after treatment with 7.5, 15 and 25 µM AQ were 61.6 ± 7.0%, 30.3 ± 6.7% and 11.0 ± 2.0% (MCF-7; Figure 3B, Control vs. AQ 7.5 µM*: p* < 0.01, Control vs. AQ 15 µM*: p* < 0.0001, Control vs. AQ 25 µM*: p* < 0.0001, AQ 7.5 µM vs. AQ 15 µM*: p* < 0.01, AQ 7.5 µM vs. AQ 25 µM*: p* < 0.001). In case of MDAMB-231, % colony growth was found to be 26.0 ± 3.0%,4.5 ± 2.7% and 0.6 ± 0.3% for AQ 7.5, 15 and 25 µM, respectively*,* considering number of colonies to be 100% in drug free treatment control wells. (Figure 3C, Control vs. AQ 7.5 µM*: p* < 0.0001, Control vs. AQ 15 µM*: p* < 0.0001, Control vs. AQ 25 µM: *p* < 0.0001, AQ 7.5 µM vs. AQ 15 µM*: p* < 0.001, AQ 7.5 µM vs. AQ 25 µM*: p* < 0.001). These data clearly representative of AQ’s efficacy in eliminating the possibility of tumor relapse, from single cancer cells left behind following chemotherapy and surgical intervention (Figure 3B,C). Data also represent the dose-dependent colony inhibition behavior of amodiaquine in two different breast cancer subtypes.

### 2.5. 3D Spheroid Studies

Efficacy of a breast cancer therapy can also be determined by assessing their ability to penetrate across solid tumors. Hence, 3D spheroid cell culture studies, mimicking the in-vivo features of tumors, were employed. In last few years, our group has established 3D tumor spheroids to be a preclinically relevant in-vitro model for testing of therapeutics and drug delivery systems in various cancers including mesothelioma, lung cancer, and breast cancer [28,31,32,33,34,35,36,37,38,39]. As discussed in the Section 4, spheroids were subjected to two kinds of dosing treatments, i.e., single and multiple dose treatments. The representative single and multiple dose spheroid images can be found in Figure 4A (MCF-7) and Figure 4B (MDAMB-231). From Figure 4A,B, it is visibly evident that AQ resulted in enhanced spheroid growth suppression compared to control throughout the experimental period. Both single and multiple dosing strategies have demonstrated the AQ’s efficacy against MCF-7 and MDAMB-231 spheroids. 

From Figure 4A, it can be found that single dose-treated spheroids exhibited a distinct tumor size reduction compared to multiple dose-treated spheroids. Amodiaquine have less solubility in buffer environment and is soluble in aqueous media. Amodiaquine precipitates when interacting with buffers. In case of multiple dose strategy, drug precipitation might have occurred due to multiple amodiaquine doses in culture media. It can be indicated that amodiaquine’s capability to internalize into cells might have been altered. Thus, amodiaquine was unable to enter cells and exert its activity. While in case of single dose strategy, media have been replaced with fresh culture media. Altogether, possibility of amodiaquine’s precipitation could have resulted in its reduced efficacy with multiple doses to MCF-7 spheroids in comparison to single dosing strategy.

All spheroids were quantified for their volume using ImageJ software (Figure 5), as discussed in Section 4. As can be seen, MCF-7 spheroids volumes were found to be 0.7 ± 0.3 mm^3^ (single dose) and 0.8 ± 0.4 mm^3^ (multiple dose) on an average on day 0 before dosing with respective treatments. It was found that spheroid volumes on day 10 after single dosing were 10.2 ± 2.0 mm^3^ (Control), 6.7 ± 1.5 mm^3^ (AQ 7.5 µM), 1.0 ± 0.5 mm^3^ (AQ 15 µM), 0.1 ± 0.3 mm^3^ (AQ 25 µM), showing dose-dependent reduction in tumor mass and viability (Figure 5A). Similarly, spheroid volumes on day 10 after multiple dosing were 12.5 ± 0.9 mm^3^ (Control), 12.6 ± 1.3 mm^3^ (AQ 7.5 µM), 5.2 ± 2.2 mm^3^ (AQ 15 µM) and 0.3 ± 0.2 mm^3^ (AQ 25 µM) (Figure 5B). 

At the same time, MDAMB-231 spheroid volumes were found to be 0.7 ± 0.3 mm^3^ (single dose) and 0.8 ± 0.4 mm^3^ (multiple dose) on an average on day 0 before dosing with respective treatments. It was found that spheroid volumes on day 10 after single dosing were 2.0 ± 0.2 mm^3^ (Control), 2.2 ± 0.1 mm^3^ (AQ 7.5 µM) and AQ 15 and 25 µM treated spheroids were completely dissipated leaving no distinct spheroid mass which represent spheroid volumes of <0.05 mm^3^ (Figure 5C). Statistically, spheroid treatment with 15 and 25 µM of AQ resulted in significant reduction in spheroid volume with single dosing as shown in Figure 5C. (Single dose: Control vs. AQ 15 µM: *p* < 0.0001, Control vs. AQ 25 µM*: p* < 0.0001). In case of multiple dosing, spheroid volumes on day 10 were found to be 1.9 ± 0.2 mm^3^ (Control), 1.8 ± 0.1 mm^3^ (AQ 7.5 µM), 0.2 ± 0.1 mm^3^ (AQ 15 µM) and <0.05 mm^3^ with AQ 25 µM treatment, as seen in Figure 5D. (Multiple dose: Control vs. AQ 15 µM*: p* < 0.0001, Control vs. AQ 25 µM*: p* < 0.0001). 

From the observed efficacy in both spheroid models, it was confirmed that treatment with AQ was able to reverse the tumor formation in an in-vitro setting, following a customizable dose-dependent manner. 

### 2.6. CellTiter-Glo Luminescent Cell Viability Assay

Measuring the extremities of the tumor/spheroid may not reveal the events at the core of the tumor, traditionally considered the hard-to-reach regions for cancer therapeutics. CellTiter-Glo luminescent cell viability assay was performed to compare counts of viable cells in treated 3D single- and multiple-dose spheroids on day 10 after microscopic quantification. As seen with volume measurement, AQ treated MCF-7 spheroids exhibited significantly reduced proportion of viable cells compared to control in a dose-dependent manner. % cell viability results are represented in Figure 6A,B for single- and multiple-dose studies of MCF-7 spheroids, respectively (single-dose: 100.0 ± 3.0%, 51.3 ± 3.8% for control and AQ 7.5 µM, ~2-fold difference; and multiple-dose: 100.0 ± 9.0% and 6.1 ± 3.1% for control and AQ 7.5 µM, 16.4-fold difference). At higher doses (15 and 25 µM), % cell viability was found to be <1% in case of both dosing strategies. In case of MDAMB-231 spheroids, significant reduction in cell viability was observed with AQ 15 and 25 µM treatment (Single dose) while treatment with 7.5, 15 and 25 µM resulted in significant reduction in % cell viability compared to control (Multiple dose) (Figure 6C,D). % cell viability results are represented in Figure 6C,D for single- and multiple-dose studies, respectively (single-dose: 100.0 ± 22.4%, 10.3 ± 7.1% for control and AQ 15 µM, 4.3-fold difference; and multiple-dose: 100.0 ± 8.0% and 23.3 ± 13.1% for control and AQ 7.5 µM, 10-fold difference). The percentage of cell viability was found to be <1% in case of multiple and single dosing strategies after treating with AQ 25 µM while ~10% cell viability was observed after single dose of AQ 15 µM. These results support the previous findings of 3D spheroid studies, that demonstrated anti-cancer effectiveness of AQ.

### 2.7. Live-Dead Cell Assay

As the physiological and 3D tumors represent biological matrices, it is imperative for the drug to penetrate through tumor’s microenvironment, in order to demonstrate its full therapeutic potential. Therefore, it is necessary to quantify dead cell and live cell portions out of a spheroid mass. Representative Live-dead cell assay images are represented in Figure 7A (MCF-7). Figure 7B (MDAMB-231). As can be seen in Figure 7, reduced green fluorescence (live cell portion) and / or increased red fluorescence (dead cell portion) indicate that AQ has resulted in spheroid cell death and diminished tumor growth. This trend was evident in case of both dosing strategies. It can be inferred that AQ have great capability in penetrating spheroid tumors.

### 2.8. Annexin V and Dead Cell Assay

The Muse Cell Analyzer determined the extent of apoptosis in MDAMB-231 cells incubated with AQ at 7.5 and 15 µM concentration for 24 h. The apoptotic profile of MDAMB-231 cells after incubating with respective treatments is shown as scatter plots in Figure 8A (representative images from *n* = 3–5 trials). Here, the right shift of the scatter plot can be clearly observed in the case of AQ (7.5 µM and 15 µM) compared to that of the control (Figure 8A). As quantification shown in Figure 8B, total % apoptotic cells were found to be 40.9 ± 2.7% (7.5 µM), 49.0 ± 1.2% (15 µM) for AQ treatment. The non-treated control exhibited 38.8 ± 2.5% total apoptotic cells. This suggests that AQ (15 µM) resulted in a statistically higher population of total apoptotic cells compared to control (*p* < 0.01). There was also a significant difference between treatments with two different concentrations of AQ (*p* < 0.01: AQ 7.5 µM vs. 15 µM). The results indicate the ability of AQ to induce early-stage apoptosis even at both concentrations of 7.5 µM and 15 µM which led to the highest proportion of total % apoptotic cells.

### 2.9. Cell Cycle Analysis

To study the mechanism of improved efficacy of amodiaquine, cell cycle analysis was conducted by measuring DNA content by propidium iodide (PI) staining method using flow cytometry. PI staining is very rapid, reproducible, and reliable method which can be used for estimation of cell cycle parameters. As can be seen in quantification presented in Figure 8C, AQ treated MCF-7 cells demonstrated a significantly reduction in % cell population in G2/M with an increase in the AQ dose, indicating the reduced cell availability for mitosis (Control: 9.3 ± 1.9%, AQ 7.5 µM: 10.1 ± 2.6%, AQ 15 µM: 8.2 ± 3.8% and AQ 25 µM: 2.5 ± 1.2%). Accompanying with this observation, an increasing trend in S phase arrest (Control: 6.7 ± 0.8%, AQ 7.5 µM: 6.6 ± 2.0%, AQ 15 µM: 11.3 ± 8.2% and AQ 25 µM: 12.4 ± 2.5%) was observed. An increasing trend was observed in % of cells in S phase (not significant) and a significantly decreasing trend of % of cells in G2/M phase (control vs. AQ 25 µM: (*p* < 0.05) and AQ 7.5 µM vs. AQ 25 µM: (*p* < 0.05)) with an increase in AQ dose was shown in Figure 8C.

### 2.10. Western Blot and Protein Expression Studies

β-catenin and LC3BII [40] play a major role in cancer progression, as previously reported [41]. β-catenin is involved in Wnt signaling pathway and its inhibition is an indication of apoptotic induction. LC3BII is an intracellular autophagy marker, and its induction is an indication of autophagy inhibition [40]. The expression levels of β-catenin, LC3B-I and LC3B-II proteins were determined using Western blot analysis, with β-actin as loading control; and the representative blots can be found in Figure 9A: MCF-7, Figure 9B: MDAMB-231) along with their densitometric quantification in Figure 9C–F. As shown in Figure 9A, AQ significantly down-regulated the levels of β-catenin and upregulated levels of LC3B-II after 24-h treatments, indicating induction of apoptosis and autophagy inhibition, respectively. These results are in accordance with the reported mechanism of action of AQ through autophagy inhibition [42]. A reduction in the β-catenin expression was observed with AQ treatments in both MCF-7 and MDAMB-231 (Figure 9C,D). Western blot analyses showed that LC3B-II expression was induced in both MCF-7 (AQ 25 µM: 14.3 ± 2.3, *p* < 0.05) and MDAMB-231 cell lines (AQ 7.5 µM: 6.7 0.2, *p* < 0.01) after AQ treatment as compared to their respective control blots (MCF-7: 1.0 ± 0.5, MDAMB-231: 1.0 ± 0.5) (Figure 9E,F).

### 2.11. Gene Expression Analysis

To further elucidate the potential mechanism for AQ’s efficacy in breast cancer therapeutics, we performed gene expression analysis in MDAMB-231 cell line after treatments with 0–20 µM concentrations of AQ. The results (Figure 10) show that AQ significantly modulated expression of a variety of genes essential for DNA transcription, and apoptotic/autophagic events. As can be seen, AQ treatment upregulated the expression of GADD34 (encodes for Growth arrest and DNA damage-inducible protein), DDIT3 (encodes for DNA Damage Inducible Transcript 3 protein), BNIP3 (encodes for BCL2 Interacting Protein 3 or BCL2/adenovirus E1B 19 kDa protein-interacting protein 3), BNIP3L (encodes for BCL2/adenovirus E1B 19 kDa protein-interacting protein 3-like), NOXA (encodes Phorbol-12-Myristate-13-Acetate-Induced Protein 1), and LC3-II (encodes for Autophagy-Related Ubiquitin-Like Modifier LC3B) (Figure 10A). These upregulated genes demonstrate the role of AQ in inducing apoptosis and autophagy related genes to act as an anti-cancer drug. In addition of upregulation, AQ also downregulated expression of several genes essential for DNA transcription such as POLR1A (encodes for RNA Polymerase I Subunit A protein), POLR1D (encodes for RNA Polymerase I and III Subunit D), POLR1E (encodes for RNA Polymerase I Subunit E), and TAF1B (encodes for TATA-Box Binding Protein Associated Factor, RNA Polymerase I Subunit B). AQ also significantly inhibited LAMP1 (encodes for Lysosome-Associated Membrane Glycoprotein 1), and CDKN1A (encodes for p21) (Figure 10B). These genes were shown to be upregulated in different cancers including breast cancer, and inhibition of the expression of these genes in a dose dependent manner by AQ treatment further confirm the anti-apoptotic potency of this drug.

## 3. Discussion

Breast cancer being the most common type of cancer among women, extensive investigations have been carried out to explore novel treatment possibilities [43]. Even though there are therapies available for treatment of multiple breast cancer subtypes\, their use is limited due to the resistance developed overtime and compromised efficacy. Hence, overall patient survival hasn’t changed [44,45]. This necessitates the development of drugs with higher efficacy, minimal side effects, and at lower cost [9]. Moreover, increasing number of new cases being diagnosed for breast cancer and the complexity in understanding the disease subtypes further demands newer treatment strategies [8,9]. As the new drug development requires a complete understanding of the appropriate targets and corresponding drugs in breast cancer treatment, the cost of the cancer treatment is rising at a higher speed [1,8]. This process is time-consuming and the use of high-end techniques is increasing the overall expenses and eventually the cost of the drug [9]. In this context, a smart development strategy is to repurpose an old, existing and FDA approved drug for a newer indication. The benefit of this approach includes the availability of information about molecular targets, mechanism of action, safety data and side effects for existing drugs [15,17]. This saves a lot of time that is often required for discovery, designing, clinical trials and approvals of a new drug while cutting down the overall costs of anti-cancer drug development [16,17].

In the current era of escalating need for new drugs against cancer, there is an emerging interest in identifying new uses for old drugs especially anti-malarial drugs [16,46]. For example, chloroquine has demonstrated promising effects as an anti-cancer agent, particularly in breast cancers. Amodiaquine (AQ) has been reported for its anti-cancer efficacy in other cancer types such as non-small cell lung cancer [31] and melanoma [29]. In the present study, we aimed to investigate the anti-tumor effects of amodiaquine in aggressive breast cancer types. 

We selected a panel of breast cancer cell lines representing major clinically relevant subtypes, responsible for resistance of this disease to chemotherapy. We profiled the activity of AQ, in terms of cell viability inhibition, cell migration and colony growth inhibition. Cell viability is an important toxicity assay parameter and is directly associated with the toxic effects of a drug [47]. AQ induced a reduction in cell viability in different breast cancer cells (MCF-7, MDAMB-231, SK-BR-3 and BT-549). The reduction was found to be dose dependent as shown in Figure 1.

Cell migration and invasion are important processes in tumor development and metastasis [48]. However, there is no evidence for the potential activity of AQ against migration in breast cancer cells. In present study, it was found that AQ dose-dependently inhibited invasiveness of MDAMB-231 cells, indicating that AQ may be a promising anti-tumor drug for treatment of breast cancer. Using a 15 or 25 µM dose of AQ, the motility of the MDA-MB-231 cells was significantly inhibited. AQ was also capable of inhibiting vascular channel formation resulting in non-existent vascular tube formations and scattered cellular geometry of MDAMB-231 cells. Clonogenic assay has been utilized to investigate the effect of amodiaquine on the colony forming ability of cancer cells, an important phenomenon for metastasis. Interestingly, clonogenic assay study revealed that AQ can inhibit the ability of single cells more efficiently to form colonies as seen in Figure 3. Hence, it ensures the effectiveness of amodiaquine on long term basis while preventing tumor recurrence. This demonstrated the ability of amodiaquine to inhibit metastasis of breast cancer cells. Overall, AQ could be further evaluated for its establishment in breast cancer treatment explicitly.

Two-dimensional in-vitro assays performed may or may not necessarily mimic in-vivo conditions due to the complexity of in-vivo tumor models and their uncontrolled cell growth, solid tumoral mass hindering the penetration of drugs and varied tumor microenvironment [49]. Thus, ex-vivo spheroid studies were performed where solid tumors were grown in specially designed ultra-low attachment 96-well plates as reported in our previous publications [22,34] to quantify the effect of amodiaquine on breast cancer. Earlier it was reported that, when cultured in the 3D systems, MCF-7 cells form spheroids, up-regulate the expression of EMT markers both at gene and protein levels. Additionally, MCF-7 and MDAMB-231 spheroids show more realistic drug responses, and provide for better evaluation of tumor proliferation and morphological changes [50]. From results seen in Figure 5, it can be understood that amodiaquine is capable of inhibiting spheroid growth inhibition. In addition, the percentage of cell viability assay results (Figure 6) on day 10 of spheroid study confirmed superior anti-cancer efficacy of amodiaquine. Hence, results from this spheroid study provide strong evidence for capability of amodiaquine to efficiently prevent tumor cell proliferation while reducing the tumor mass. In agreement with earlier observations in the current study, amodiaquine was found to be effective in suppressing tumor growth significantly as compared to control in both single and multiple treatment methods. In addition, live-dead cell assay infers that reduced live cell population was observed with AQ treatment. Therefore, it can be suggested that amodiaquine is a promising molecule, which is proven to be effective in producing desired therapeutic outcomes. Thus, based on demonstrated efficacy of AQ in 3D spheroid models which are capable of mimicking the *in-vivo* tumor conditions, it could be further tested in preclinical studies. 

Furthermore, the AQ was further explored to illustrate its mechanism of action through estimation of cell cycle parameters using flow cytometry. It has been observed that amodiaquine is capable of causing cell arrest in S-phase, hence hindering them from entering mitotic phase as can be seen in Figure 8C. Post G2-M phase, the cells differentiate into daughter cells and undergo mitosis. Plot representing cell cycle reveals that population of AQ-treated cells in G2-M phase is significantly less than control cells, representing lesser cell division after AQ treatment. Autophagy is a catabolic biological event characterized by the degradation of the cellular compartments and their recycling in order to improve cell survival upon harsh living environment [51]. Microtubule-associated protein light chain 3B-II (LC3B-II) performs the key processes of each phase of autophagy [52]. LC3B-II is a quantitative marker of autophagy since it is required for the formation of the autophagosome and its level is proportional to the amount of autophagosomes in the cells [53]. Apoptosis was also evaluated by assaying for β-catenin, a general marker of apoptosis [54]. Immunoblotting analysis demonstrated the dramatic decrease in the levels of β-catenin and increased levels of LC3B-II as can be found in Figure 9C–F in both breast cancer subtypes. Similarly, we observed overexpression of some critical cancer related and regulatory genes (Figure 10A) such as GADD34, DDIT4, BNIP3, BNIP3L, NOXA, and LC3B-II in response to AQ treatments confirming the role of AQ in regulating the autophagy and cancer regulatory genes [55,56,57,58]. Additionally, we observed downregulation of genes such as POLR1A, POLR1D, POLR1E, TAF1B, LAMP1, and CDKN1A (Figure 10B) in response to AQ treatment demonstrating the role of AQ in regulating RNA polymerase subunits and effect on overall gene transcription in breast cancer [58,59,60]. The inhibition of CDKN1A that encodes for p21, a cell cycle regulator further validate our results AQ mediated cell cycle arrest, similar type of observations were made earlier for different cancers including breast cancer [60].

To our knowledge, this is the first study reporting the efficacy and detailed molecular mechanism of action of amodiaquine for treatment of breast cancer. Taken together, supporting evidence from various in-vitro and ex-vivo studies demonstrated the superior anti-tumor activity of amodiaquine for the treatment of breast cancer.

## 4. Materials and Methods

### 4.1. Cell Lines and Materials

MCF-7 (HER2 positive), MDAMB-231 (TNBC), BT-549 (TNBC), and SK-BR-3 (HER-2 positive) breast cancer cell lines were obtained from ATCC (Manassas, VA, USA). MCF-7, and MDAMB-231 were maintained in DMEM medium (Corning, NY, USA) supplemented with 10% FBS (Atlanta Biologicals, Minneapolis, MN, USA) and penicillin-streptomycin (Corning, NY, USA) at 5% CO_2_/37 °C. BT-549 was maintained in RPMI-1640 medium supplemented with 10% FBS (Atlanta Biologicals, Minneapolis, MN, USA), 1% sodium pyruvate (Corning, NY, USA), 1% penicillin-streptomycin (Corning, NY, USA), 0.023 U/mL of Gibco™ Insulin, human recombinant zinc solution (Fisher Scientific, Hampton, NH, USA)). SK-BR-3 was maintained in Hyclone McCoy’s 5A Medium (GE Health care Life Sciences (Marlborough, MA, USA)) supplemented with 10% FBS (Atlanta Biologicals, Minneapolis, MN, USA) and 1% penicillin-streptomycin (Corning, NY, USA). 3-(4,5-dimethylthiazol-2-yl)-2,5-diphenyltetrazolium bromide (MTT), dimethyl sulfoxide (DMSO), crystal violet dye, and 16% paraformaldehyde (PFA) solution were purchased from Fisher Scientific (Hampton, NH, USA). All molecular biology kits and supplies were purchased from other commercial vendors which are listed at appropriate places throughout the manuscript. 

### 4.2. Methods

#### 4.2.1. Cytotoxicity Studies

AQ was evaluated for its cytotoxicity efficacy in four different breast cancer cell lines: MCF-7, MDAMB-231, SK-BR3, and BT-549 as reported earlier with slight modifications [22,32]. Detailed methods are provided in Appendix A. 

#### 4.2.2. Scratch Assay

In-vitro scratch assay was used to study the cell migration. Briefly, scratches were created on a confluent cell monolayer. The cells on the edge of the scratch will migrate toward the center to close the scratch, thus establishing new cell-cell contacts. The assay was performed on MDAMB-231 cell line as previously reported [33,61]. Detailed methods are provided in Appendix A. 

#### 4.2.3. Vasculogenic Mimicry Assay

Vasculogenic mimicry (VM) refers to the unique capability of aggressive tumor cells to mimic the pattern of embryonic vasculogenic networks. Recent studies have found that some highly aggressive tumor cells generate vessel-like channels in the absence of endothelial cells or fibroblasts [62,63]. These channels are thought to provide a new mechanism of perfusion and a dissemination route within the tumor. Previous studies have demonstrated with VM’s association with more aggressive tumor phenotype and poor prognosis in patients [64]. To understand AQ’s ability to inhibit VM phenomena, VM assay was performed in MDA-MB231 cell line, as per the previously published methods [64,65] from our research group with slight modifications. Briefly, 50 µL of Cultrex^®^ RGF BME (R&D Systems; Minneapolis, MN, USA) was added to each well of a 96-well plate and incubated at 37 °C for 60 min to allow the BME to gel. Briefly, MDAMB-231 cells were seeded at 2.0 × 10^4^ cells/well on this BME-coated 96-well plate after preparing dilutions with varying concentrations (7.5, 15 and 25 µM) of AQ. Cells were incubated for 8 h at 37 °C/5% CO_2_. Phase contrast images were taken using an inverted microscope (Laxco, Mill Creek, WA, USA) with 10× magnification, and effect of AQ on tube formation was qualitatively observed from the images.

#### 4.2.4. Clonogenic Assay

Using this assay, the effectiveness of the AQ towards colony inhibition was determined in MCF-7 or MDAMB-231 cells. Protocols reported previously [31,66] was briefly modified and followed in this study. Detailed methods are provided in Appendix A. 

#### 4.2.5. 3D Spheroid Study

Compared with two-dimensional (2D) monolayer culture, breast cancer spheroids more accurately reflect the complex microenvironment in-vivo [50]. Many of our recent studies have also reported ability of 3D spheroid culture to mimic the in-vivo features of tumors [32,35]. Spheroid models are especially efficient in detecting malignant cells and tumorigenesis while also assessing drug resistance. According to several studies, the breast cancer cell lines cultured in 3D spheroids have superior ability to mimic 3D assembly of cancerous tissue while being relevant to tumor microenvironment [67,68]. For instance, the BT-474 spheroids modulate the distribution of human epidermal growth factor receptor-2 (HER2), and show a higher anti-apoptotic level than those cultured in 2D monolayer [69,70]. MCF-7 spheroids reveal the role of tumor microenvironment in metastasis, and are more resistant to drug treatments than those cultured as a 2D monolayer [71]. For this study, a 3D cell-based spheroid model was developed for MCF-7 and MDAMB-231 cell lines, following previously established methods [35] Detailed methods are provided in Appendix A. 

#### 4.2.6. CellTiter-Glo 3D Luminescent Cell Viability Assay

CellTiter-Glo cell viability assay was performed using a commercially available kit (CellTiter-Glo^®^, Promega, Madison, WI, USA) on MCF-7 and MDAMB-231 spheroids on 10th day according to manufacturer’s protocol. After imaging (10× magnification), 100 µL of medium was removed and replaced with 100 µL of CellTiter-Glo^®^ reagent in each well. The contents were mixed for 2 min, followed by 30 min incubation at room temperature. The luminescence was measured using Spark 10 M plate reader (Tecan, Männedorf, Switzerland). The results were presented as % cell viability (mean ± SD; *n* = 3) and compared with fresh media blank controls.

#### 4.2.7. Live-Dead Cell Assay

Live & Dead cell assay was performed using viability/cytotoxicity assay kit for Animal Live & Dead Cells (Biotium, Fremont, CA, USA). According to manufacturer’s protocol, live-dead cell study was performed on treated MCF-7 and MDAMB-231 spheroids on day 10th of both single and multiple dosing in the therapeutic model. Briefly, after complete removal of media from the wells; 100 μL of staining solution (2 μM calcein AM/4 μM Ethidium homodimer III (EthD-III)) was added to treated spheroids; in order to give the green/red fluorescent staining for viable and dead cells, respectively. Plate was then incubated at room temperature for 45 min in dark. Images were captured at 4× magnification using fluorescence microscope (EVOS-FL, Thermo Fisher Scientific, Waltham, MA, USA). 

#### 4.2.8. Muse Annexin V and Dead Cell Assay

Apoptotic cell distribution of MDAMB-231 cells was assayed by using the MUSE Annexin V & Dead Cell Kit (Millipore, Billerica, MA, USA) according to the manufacturer’s instructions. Briefly, MDAMB-231 cells were seeded (100,000 cells/well) in a 24 well plate and incubated overnight. The next day, the media was replaced with AQ (25, 15, 7.5 µM), or fresh media. After 24 h, treatments were removed, and cells were collected through trypsinization followed by washing with PBS twice. Cell suspension was diluted with growth media to a concentration of 0.5 × 10^6^ cells/mL; and 150 µL of Annexin V/dead reagent and 100 µL of a single cell suspension were mixed in a microtube thoroughly by vortexing for 5 s, followed by incubating in the dark for 20 min at room temperature. Cells were then analyzed using the Muse cell analyzer (Luminex, Austin, TX, USA). The apoptotic ratio was determined by identification of four populations: (i) non-apoptotic cells, not undergoing detectable apoptosis, Annexin V (−) and 7-AAD (−); (ii) early apoptotic cells, Annexin V (+) and 7-AAD (−); (iii) late apoptotic cells, Annexin V (+) and 7-AAD (+); and (iv) cells that have died through non-apoptotic pathway, Annexin V (−) and 7-AAD (+).

#### 4.2.9. Cell Cycle Analysis

Following treatment for 72 h time period, cell cycle was studied using propidium iodide (PI) assay with flow cytometry using reported protocols with slight modifications [72]. Briefly, MCF-7 cells were seeded in tissue culture treated 6 well plate at a density of 100,000 cells/well and incubated with AQ: 7.5, 15 and 25 μM for 72 h. After treatment, the cells were collected by trypsinization followed by washing two times with PBS. After washing, cells were fixed in cold 70% ethanol for 1 h followed by 2× washing with PBS. Thereafter, cells were incubated with RNA-ase (100 µg/mL) and PI (10 µg/mL) for 30 min in dark at room temperature. Cells (50,000 counts/sample) were analyzed for PI signals using flow cytometer Flowsight Amnis Cor. (Luminex, Austin, TX, USA) and data were processed using IDEAS^®^ software.

#### 4.2.10. Western Blot Analysis

MCF-7 and MDAMB-231 cells were plated 1 × 10^6^ cells per petri dish and were treated with AQ (7.5, 15 and 25 μM: 24 h) at 37 °C/5% CO_2_. After treatment, the cells were collected and lysed with 1% Triton^®^ X-100 (Fisher Bio-Reagents, Hampton, NH, USA) and 1% Halt™ Protease and Phosphatase Inhibitor Cocktail (Thermo-Fisher Scientific, Waltham, MA, USA) in PBS and bath sonicated for 1 h at 4 °C. Samples were centrifuged for 15 min at 4°C at 15,000 rpm and lysates were collected. Cell lysate protein was quantified by DC™ Protein Assay Kit (Bio-Rad, Hercules, CA, USA). The samples were mixed with 2× Laemmli buffer Bio-Rad, Hercules, CA, USA) and 2-mercaptoethanol, and denatured at 110 °C for 10 min. For Western blot analysis, 10 µg protein was loaded and separated on 4–20% Mini- PROTEAN^®^ TGX™ Precast Protein Gels (Bio-Rad, Hercules, CA, USA) and transferred to Trans-Blot^®^ Turbo™ Midi PVDF membranes (Bio-Rad, Hercules, CA, USA) using a Bio-Rad PowerPac™ Basic Power Supply and Trans-Blot^®^ Turbo^TM^ system. The membranes were blocked with 5% bovine serum albumin in PBS and probed with corresponding primary antibodies (1:1000 dilution) and kept for overnight shaking at 4 °C. The following antibodies were used: β-actin (PA1-183, Invitrogen, Waltham, MA, USA); LC3B-II (Cell Signaling Technology, Beverly, MA, USA) and β-catenin (13-8400, Invitrogen). Membranes were then incubated with corresponding secondary HRP-conjugated antibodies: goat anti-mouse (#31430) poly-HRP (1:10,000 dilution) and goat anti-rabbit (#32260) (1:10,000 dilution) (Thermo Fisher Scientific, Waltham, MA, USA) for 1 h at room temperature and subjected to Western Bright chemiluminescence (WBF25, The Gel Company, San Francisco, CA, USA). Protein signals were detected on the membranes and were quantified using the chemiluminescent imaging by the Omega Lum™ G Imaging System (The Gel Company, San Francisco, CA, USA).

#### 4.2.11. RNA Extraction

Triple-negative breast cancer cell line MDAMB-231 was used for real-time quantitative polymerase chain reaction (RT-qPCR) experiments. 1 million cells/well were seeded in a 6-well plate as described above followed by culturing for 24 h. Confluent (80%) cells were treated with different concentrations of amodiaquine (0–20 µM) for 12 h in duplicates. Following the drug treatment, cells were trypsinized, washed twice with ice-cold PBS, and cell pellets were collected. Total RNA was extracted from cells using the Qiagen RNeasy plus mini kit (Qiagen, Germantown, MD, USA) according to the manufacturer’s instructions. Briefly, cell pellets were homogenized using RLT buffer, and cell lysates were passed through a gDNA elimination column to remove any genomic DNA from the lysate. RNA was precipitated using 70% ethanol treatment of the lysate followed by binding on the RNeasy mini column. Column bound RNA was washed once with buffer RW1 followed by washing twice with buffer RPE before eluting with water. The eluted RNA was quantified using a Nanodrop spectrophotometer (Thermo Fisher Scientific, Waltham, MA, USA), and purity of RNA was determined by the absorbance ratio of 260/280 nm. 

#### 4.2.12. cDNA Synthesis

Equal amount of total RNA was used to synthesize single stranded cDNA using High-Capacity cDNA Reverse Transcription Kit (Thermo Fisher Scientific; 4368814). Briefly, RT buffer (10×), 100 nM dNTP mix, RT random primers (10×), RNase inhibitor, and MultiScribe Reverse Transcriptase were used in a PCR reaction to convert RNA to cDNA. PCR reactions were carried out using the MiniAmp Thermal cycler (Thermo Fisher Scientific, Waltham, MA, USA) at standard cDNA amplification cycle. 

#### 4.2.13. Gene Expression Analysis

RT-qPCR reactions were performed using the SYBR Green dye technology (Thermo Fisher Scientific; 4385610) to determine the gene expression changes in response to drug treatments. Specific gene specific primers were designed (Appendix A) and used with SYBR Green dye and cDNA. The reactions were carried out using the QuantStudio 3 Real-Time PCR System (Thermo Fisher Scientific, Waltham, MA, USA) and the gene expression levels were determined using the GAPDH as a housekeeping gene. All RT-qPCR reactions were performed in triplicated and repeated twice. The p values were calculated by Student’s *t*-test for expression fold difference of individual genes. 

#### 4.2.14. Data Analysis and Statistical Evaluation

All data were addressed as mean ± SD or SEM, with *n* = 3 unless otherwise mentioned. Three trials of cytotoxicity studies were performed for each control or treatment with *n* = 6 for each trial. All data were evaluated by unpaired student’s *t*-test or one-way ANOVA followed by *Tukey’s* multiple comparisons test, using GraphPad Prism software (Version 9.01 for Windows, GraphPad Software, CA, USA).

## 5. Conclusions

Research targeting finding new anti-cancer therapies is prompted by cancers’ high mortality rate. Anti-malarial drugs have taken their place in the research arena as new, effective medicines in treating various breast cancer sub types. The current study establishes the overall utility of an antimalarial drug, AQ, in treatment of breast cancer by evaluating its anticancer activities through several in-vitro cell culture studies as well as 3D spheroid models. Altogether, AQ has demonstrated superior anti-tumor properties illustrated by its potential to inhibit autophagy, induce apoptosis, and cause cell cycle arrest. Even though animal studies were not employed to assess AQ’s efficacy, recent studies have compared 3D spheroids and in-vivo studies, and have reported similar outcomes in both cases [73,74]. Furthermore, spheroid culture demonstrates metabolic resemblances to the original tissue and in-vivo experiments; and is being broadly used to replace animal experiment, not just for cancer but also for other diseases [75]. While efficacy of AQ has been proven in different in-vitro models together with in-vivo cancer microenvironment simulating 3D spheroid studies, preclinical proof of concept in-vivo studies is needed to embrace the full spectrum of AQ repurposing approach’s feasibility. As we develop optimal dosage form for AQ to be used in breast cancer treatment, optimal preclinical studies will be performed. 

## Figures and Tables

**Figure 1 ijms-23-11455-f001:**
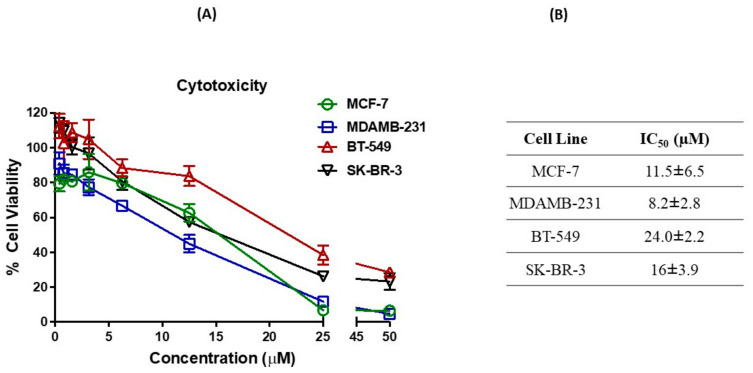
(**A**) Inhibitory effects on different breast cancer cell lines (MCF-7, MDAMB-231, BT-549 and SK-BR-3) after treatments with AQ. (**B**) IC_50_ of AQ in MCF-7, MDAMB-231, BT-549 and SK-BR-3. Data represent mean ± SD (*n* = 6) of at least 3 independent trials.

**Figure 2 ijms-23-11455-f002:**
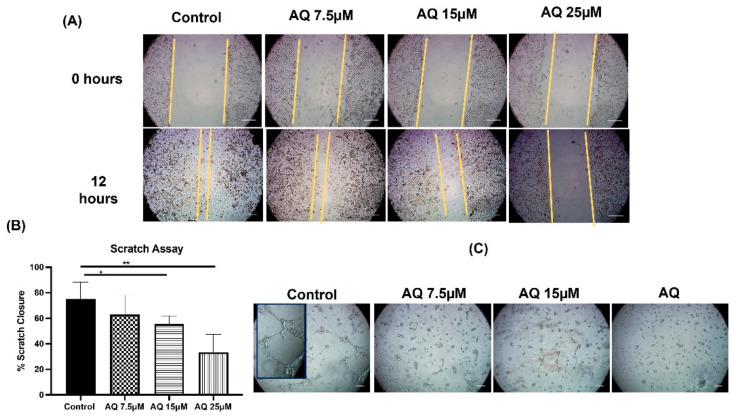
Scratch assay: In vitro scratch wound healing assay with MDAMB-231 cells treated with AQ (7.5, 15 and 25 µM) with no treatment as a control. (**A**) Shows representative images for indicated treatments (**B**) The graph shows the percent area closure in scratch assay after 12 h. Significance between the groups was analyzed by unpaired student’s *t*-test. Scale Bar 500 μm. Data represent mean ± SD (*n* = 4). (**C**) Effect of AQ on vasculogenic mimicry.; Inset image in control group represents the closer view of the tubular network.; scale bar represents 500 μm. * *p* < 0.05, ** *p* < 0.01.

**Figure 3 ijms-23-11455-f003:**
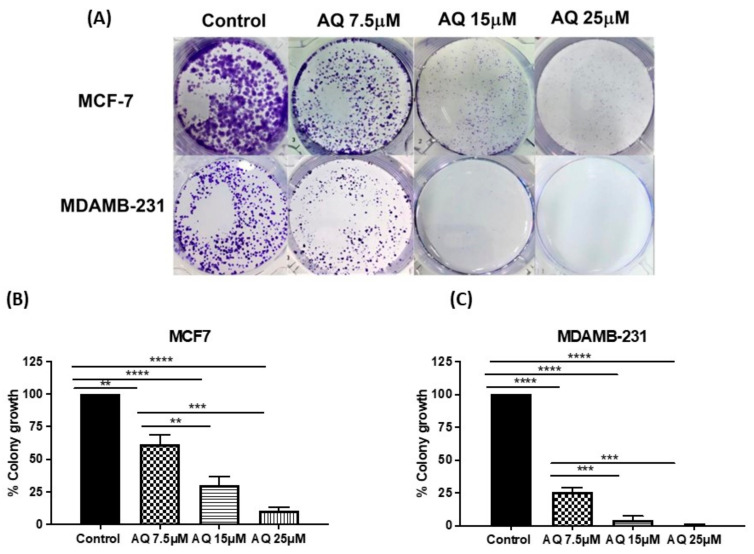
(**A**) Representative images showing distinct colonies after staining in MCF-7 and MDAMB-231 cell lines. Three different experiments were performed. (**B**,**C**) Quantitative representation of clonogenic assay as % colony growth with AQ’s treatment as compared to control in MCF-7 and MDAMB-231 cell lines, respectively. Data represent mean ± SEM (*n* = 3). ** *p* < 0.01, *** *p* < 0.001, **** *p* < 0.0001.

**Figure 4 ijms-23-11455-f004:**
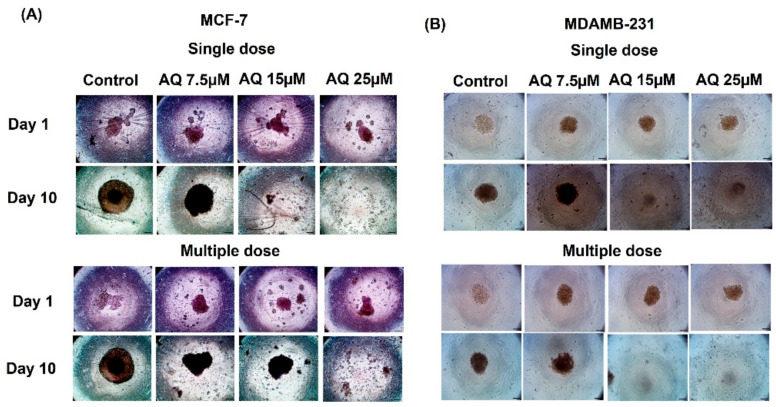
3D Spheroid study: (**A**) Representative spheroid images of MCF-7 cell line (Single dose and Multiple dose) (**B**) Representative spheroid images of MDAMB-231 cell line (Single dose and Multiple dose).

**Figure 5 ijms-23-11455-f005:**
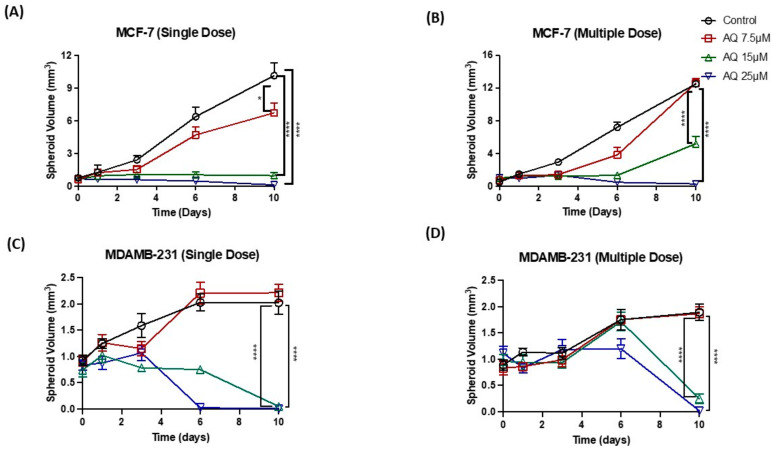
(**A**,**B**) Spheroid volume comparison plots for MCF-7 cell line. (**A**) Single dose. (**B**) Multiple dose. (**C**,**D**) Spheroid volume comparison plots for MCF-7 cell line. (**C**) Single dose. (**D**) Multiple dose. Data represent mean ± SEM (*n* = 6). * *p* < 0.05, **** *p* < 0.0001.

**Figure 6 ijms-23-11455-f006:**
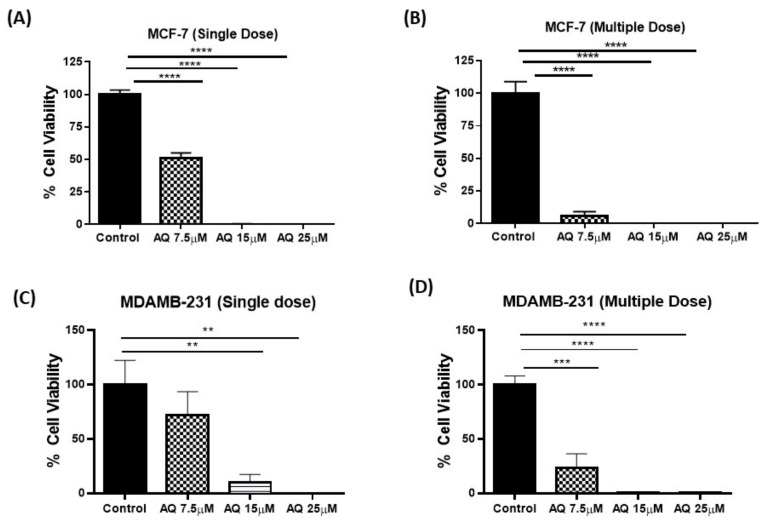
(**A**,**B**) Cell viability study on MCF-7 Spheroids. (**C**,**D**) Cell viability study on MDAMB-231 Spheroids. The results indicate % cell viability after each treatment, comparisons were made by considering control as 100%. Data represent mean ± SD (*n* = 3). ** *p* < 0.01, *** *p* < 0.001, **** *p* < 0.0001.

**Figure 7 ijms-23-11455-f007:**
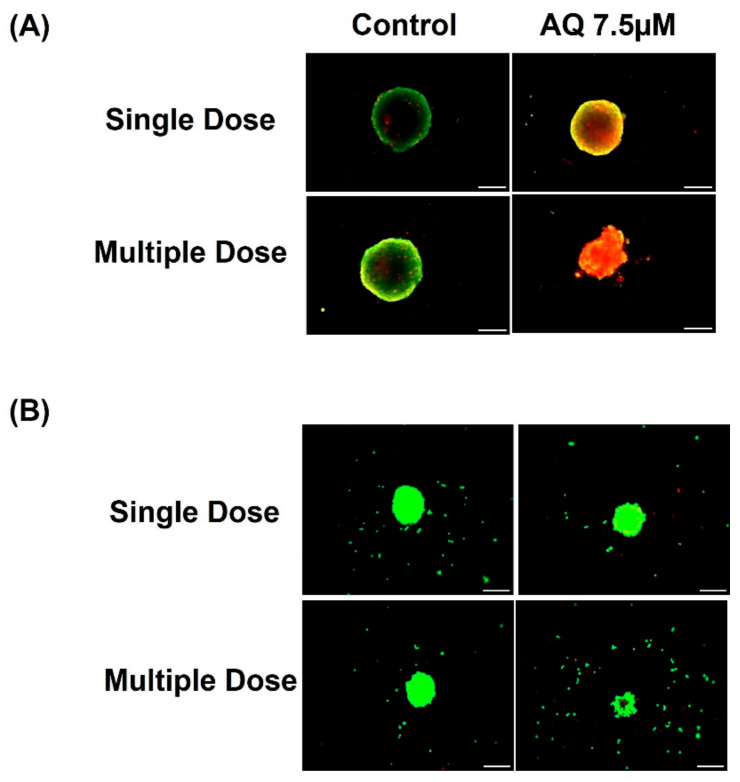
Representative live-dead stained spheroid images (**A**) MCF-7 and (**B**) MDAMB-231.

**Figure 8 ijms-23-11455-f008:**
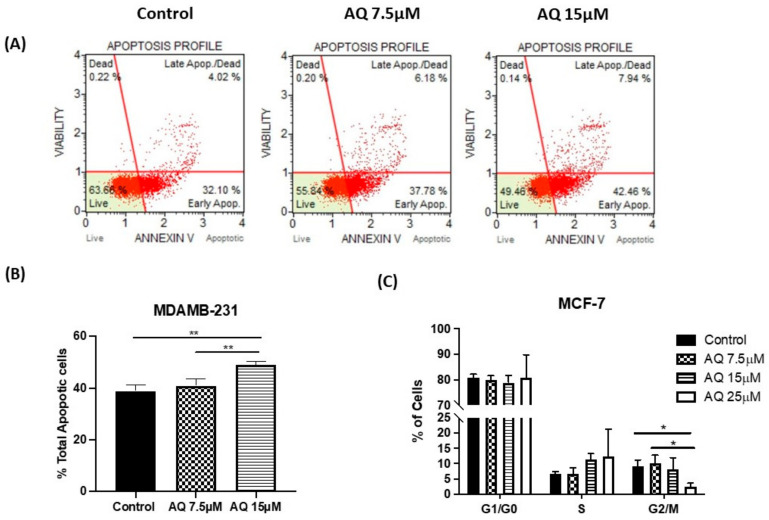
(**A**) Apoptotic impacts of different concentrations of AQ on MDAMB-231 cell line. (**B**) Total % apoptotic profile: Graph representing total % of apoptosis when treated with AQ (7.5 and 15 µM) relative to control. (**C**) Plot represents population of cells in various phases of cell cycle. A significant difference can be seen in the population of cells in the G2/M phase as AQ concentration is increased from 7.5 to 25 µM, indicating less population of cells are pushed towards mitosis. Data represents mean ± SD (*n* = 3), * *p* < 0.05, ** *p* < 0.01. compared between treatment and control group and among treatment groups as indicated.

**Figure 9 ijms-23-11455-f009:**
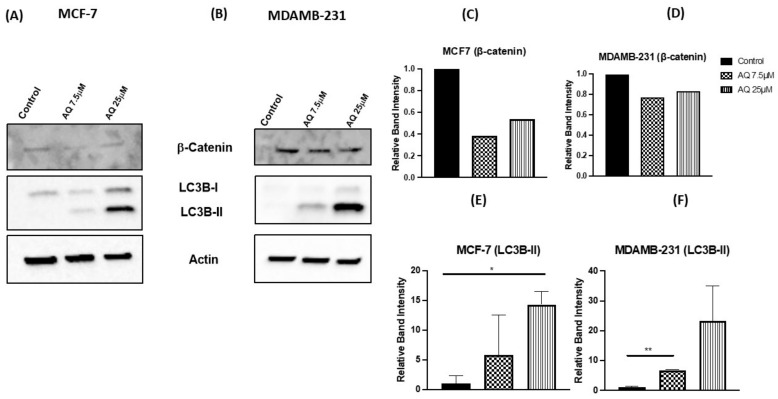
Western blots representing inducing effect of treatments on expression of β-catenin, and LC3BII proteins in MCF-7 (**A**) and MDAMB-231(**B**) cells. (Number of experimental trials: LC3B-II: *n* = 2; β-catenin: *n* = 1). (**C**,**D**) Western blot representing expression level of β-catenin in MCF-7 and MDAMB-231 cell lines, respectively. (**E**,**F**) Western blot representing expression level of LC3B-II in MCF-7 and MDAMB-231 cell lines, respectively. * *p* < 0.05, ** *p* < 0.01.

**Figure 10 ijms-23-11455-f010:**
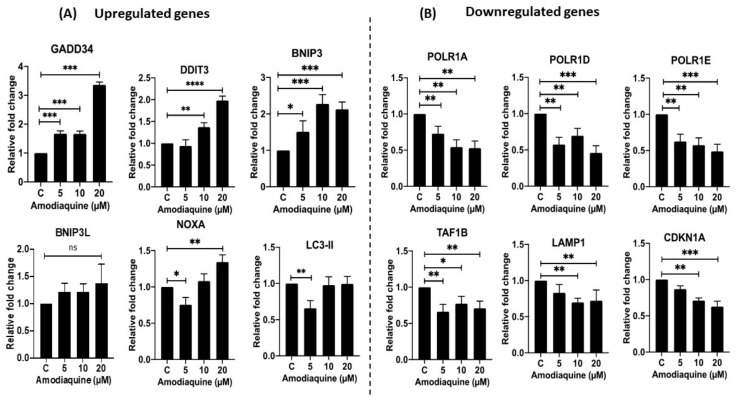
Gene expression analysis by RT-qPCR showing (**A**) upregulated and (**B**) downregulated genes in response to different amodiaquine concentrations in MDAMB-231 cells. Data for individual genes is represented as mean ± SD (*n* = 3), * *p* < 0.05, ** *p* < 0.01, *** *p* < 0.001, **** *p* < 0.0001, compared between treatment and control group and among treatment groups as indicated.

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
