# Peer review of "Exploring Amodiaquine’s Repurposing Potential in Breast Cancer Treatment—Assessment of In-Vitro Efficacy & Mechanism of Action"

_ijms, 2022, doi:10.3390/ijms231911455_

Round 1

Reviewer 1 Report

In this manuscript, the authors tested the anti-cancer effect of amodiaquine (AQ) using different breast cancer cell lines. Although various data are presented, the novelty is limited because the anti-cancer effect of AQ has been evaluated on other cancer types. Furthermore, some important issues are not dissolved. For example, after administration, the AQ will be converted into another structure by liver, will this affect its anticancer effect? In vivo anti-cancer study should be performed to demonstrate the application potential of AQ.

Reviewer 2 Report

 The authors need to address the following issues before I can recommend its publication: 

1.        “AQ may have the capability to exert its anti-cancer efficacy via Estrogen and progesterone receptors predominantly …” but MDA-MB-231 cell lines not expressing any of these ER/PR/HER2 receptors.

2.       Vasculogenic Mimicry Assay, tubular network is not clear in control group, the network should consist of many small grids. Additionally, cells should stain with Calcein AM Assay to confirm the cells were alive.

3.       In “3.5. D Spheroid Studies” section, the discussions are not enough. For MCF-7 cells, why the AQ’s efficacy in multiple dose is lower than single dose.

4.       In Fig4 MDA-MB-231, AQ 15uM, Day 10, the spheroid is clear on the left of the picture, but in Fig 5, spheroid volume is nearly 0, please give explanation.

5.       The description in line 482 and 483 is not properly, it is green fluorescence not protein.

6.       In Fig. 7, why is there red fluorescence in MCF 7 but no red fluorescence (Dead cell portion) in MDA-MB-231 cells.

7.      The title is “Repurposing an Antimalarial Drug for Efficacy in Diverse Breast Cancer Subtypes”, but the main experiments only used MCF-7 and MDAMB-321 breast cancer cell lines, and other breast cancer subtypes need to be added to this article.

8.       Scale bars should be indicated in Figure 4 and 7;

9.       Some related references should be added: Exploration. 2022, 2, 20210238.

Round 2

Reviewer 1 Report

The manuscript has been improved and can be published.